# Cryopreserving Rabbit Semen: Impact of Varying Sperm Concentrations on Quality and the Standardization of Protocol

**DOI:** 10.3390/vetsci11010009

**Published:** 2023-12-22

**Authors:** Michele Di Iorio, Fabrizio Lauriola, Giusy Rusco, Emanuele Antenucci, Michele Schiavitto, Nicolaia Iaffaldano

**Affiliations:** 1Department of Agricultural, Environmental and Food Sciences, University of Molise, Via De Sanctis snc, 86100 Campobasso, Italy; michele.diiorio@unimol.it (M.D.I.); f.lauriola2@studenti.unimol.it (F.L.); giusy.rusco@unimol.it (G.R.); e.antenucci@studenti.unimol.it (E.A.); 2Italian Rabbit Breeders Association (ANCI-AIA), Volturara Appula, 71030 Foggia, Italy; micheleschiavitto@anci-aia.it

**Keywords:** rabbit semen, sperm concentrations, standardizing freezing protocol, motility parameters

## Abstract

**Simple Summary:**

In the rabbit species, the use of frozen semen could play a twofold role; on the one hand, it could bring many benefits to commercial breeding operations that rely on artificial insemination, and on the other hand, it would enable the conservation of biodiversity through the establishment of semen cryobanks. One of the less explored aspects of cryopreserving rabbit semen involves establishing the ideal sperm concentration in straws. Achieving this goal is crucial for minimizing the variability in outcomes and accurately determining the number of sperm delivered during artificial insemination procedures. This study sought to determine the ideal sperm concentration in straws for cryopreserving rabbit semen, with the broader goal of improving and standardizing the freezing protocol developed by our research group over the past decade. Our study provided a comprehensive analysis of how different sperm concentrations within straws (ranging from 15 to 75 million sperm) influenced the critical stages of the cryopreservation process, starting from initial dilution with the extender, through to cooling and equilibration, and culminating in the thawing phase. This investigation shed light on the role of the sperm concentration when determining the overall success of rabbit semen cryopreservation.

**Abstract:**

This study aimed to investigate the impact of sperm concentrations on the in vitro quality of cryopreserved rabbit semen. The semen pools (*n* = 8, from 80 donors) were split into five aliquots with final sperm concentrations of 15, 25, 35, 55, and 75 × 10^6^ per straw. The sperm motility parameters (CASA system) and membrane integrity (flow cytometric analysis) were both evaluated at various stages of the cryopreservation process: fresh semen dilution, cooling, equilibration, and immediately after and 30 min post-thawing. The results indicated the significant influence of the sperm concentration on the total motility (TM) and progressive motility (PM), with a consistent decline in all sperm variables over the time points. Notably, the semen with a final concentration of 15 × 10^6^ exhibited a higher TM and PM after cooling and equilibration. The post-thawing quality (TM, PM) was higher (*p* < 0.05) in the mid-range sperm concentrations of 25 × 10^6^ (49.9% and 19.7%) and 35 × 10^6^ (46.2% and 19.7%) compared to the other concentrations. This study demonstrated that the sperm concentration per straw played a significant role in specific phases of the cryopreservation process. These findings contribute valuable insights for refining and standardizing the cryopreservation protocol for rabbit semen, emphasizing the importance of the sperm concentration.

## 1. Introduction

The limited survival rate of rabbit sperm following cryopreservation represents a significant challenge that hinders the widespread adoption of frozen semen in artificial insemination (AI) programs. It also poses as a barrier to the effective preservation of genetic resources through the establishment of a sperm cryobank.

Numerous studies have explored ways to enhance the cryopreservation of rabbit sperm by addressing aspects such as freezing techniques and the types and concentrations of cryoprotectants [1,2,3]. Although significant technical improvements have been made to the protocols used on rabbit semen in recent years [3,4], the absence of a well-established standard freezing procedure in this species is due to the inconsistent and varied outcomes observed in different studies. Currently, post-thaw results show variable sperm survival rates, averaging between 10% and 60% for total motility and viability and 5% to 35% for progressive motility. These fluctuations emphasize the urgent requirement to enhance and standardize the cryopreservation protocols for rabbit semen [1,2,3,4].

One contributing factor to the inconsistent results in rabbit sperm cryopreservation may stem from variations in sperm quality [4]. An important aspect that contributes to the variability of the results obtained is the unpredictability of the final sperm concentration within the straws, as freezing protocols often rely on diluting the semen with freezing media at fixed dilution ratios. Moreover, it is recognized that the freezing success of rabbit sperm varies among different breeds [5], and also among donors [6].

In the last decade, we have extensively investigated various aspects of rabbit semen freezing protocols [7,8,9,10], and recently, to fully understand the molecular mechanisms that cause damage during cryopreservation, we have studied the proteome of fresh and frozen rabbit semen to identify the proteins that are altered during cryopreservation and may affect the post-thaw sperm quality [11]. Combining the findings from these studies, we devised an optimal semen freezing protocol, which involved cooling the sperm at 5 °C for 90 min, diluting it with a freezing extender (tris–citric acid–glucose—TCG) containing 16% dimethylsulfoxide (DMSO) and 0.1 M of sucrose, 45 min equilibration at 5 °C, and exposing it to liquid nitrogen vapor at 5 cm above the nitrogen. This protocol enabled us to achieve reproductive outcomes with frozen semen that were comparable to those observed with fresh semen [7,8,9,10]. However, in our previous works, using the semen pool to balance the variability in sperm concentration (ranging from a minimum of 100 million to a maximum of 900/1000 million/mL) and the fixed semen extender dilution rate in the freezing protocol (1:1), the concentration per straw ranged between 40 and 75 million spermatozoa, which is one of the major drawbacks and limitations of our protocol. In the current study, we opted to examine lower concentrations (15, 25, and 35 × 10^6^) to align with the prospect of freezing semen from individual donors for the Italian rabbit semen cryobank. Individual donors in these breeds may exhibit lower sperm concentrations. Moreover, the inclusion of these concentrations aims to explore a broader range, recognizing the significance of evaluating lower sperm concentrations.

Therefore, the determination of an accurate sperm concentration in the straws is necessary for the standardization of rabbit cryopreservation protocols, to reduce the variability in the results and to improve upon the current freezing protocols. In this context, the dairy bull industry represents a role model, as the protocols for freezing bull sperm are highly standardized and include the utilization of specified and consistent sperm concentrations in each straw [12].

In this study, in order to find a standardized freezing protocol for this species, our main goal was to assess the impact of varying the sperm concentrations within straws (15, 25, 35, 55 and 75 × 10^6^) on the quality of cryopreserved semen by evaluating the sperm motility parameters and sperm membrane integrity. A comprehensive investigation of evaluating the effects of sperm concentration in critical steps of the cryopreservation procedure, spanning from initial dilution with the extender through to cooling, equilibration, and ultimately thawing, was undertaken. This elucidated our understanding of the role of the sperm concentration in the overall success of rabbit semen cryopreservation.

## 2. Materials and Methods

### 2.1. Chemicals

All the chemicals used in the study were of the highest commercial purity available. Unless otherwise stated, all chemicals were purchased from Merck (Milan, Italy).

### 2.2. Animals

In this research, 80 adult rabbit bucks (7–9 months old) of the Bianca Italiana breed were used. The animals were kept at the Central Breeding Farm of the Italian Rabbit Breeders Association (ANCI-AIA, Volturara Appula (FG), Italy). Throughout the study, the rabbits were individually housed in flat-deck cages and provided with a 16 h light/8 h dark photoperiod. A commercial standard breeder diet and free access to water were provided.

### 2.3. Semen Collection and Macroscopic Evaluation

The ejaculates were collected using a teaser doe and a pre-heated artificial vagina. Only ejaculates that exhibited a homogeneous white opalescent color were used in the experiment, while samples containing urine and/or cell debris were discarded. When present, gel plugs were removed immediately after collection using Pasteur pipettes. Ejaculates were pooled (9–11 ejaculates/pool, final volume of 6.5 mL) to avoid the effects of individual differences among males; in total, eight pooled semen samples were used.

### 2.4. Semen Processing and Experimental Design

The semen samples were transferred from the farm to the laboratory within 30 min, and they were held in a polystyrene box at a temperature of around 30 °C. Once in the laboratory, a semen aliquot was immediately taken from each pool for the evaluation of the fresh semen quality (sperm motility, membrane integrity, and sperm concentration).

For each fresh semen sample pool, the sperm concentration was assessed using spectrophotometric analysis. This involved measuring the optical density at 530 nm of samples diluted 1:200 in a 0.9% NaCl solution. The sperm concentration was calculated using interpolation on a previously calibrated calibration curve and expressed in millions/mL.

Each pool was subsequently divided into five aliquots, and each one was initially diluted with tris–citrate–glucose (TCG; 250 mmol/L tris-hydroxymethylaminomethane, 88 mmol/L citric acid, and 47 mmol/L glucose, pH 6.8) and subsequently with TCG supplemented with cryoprotectants (CPAs) (freezing extender) until it reached the predetermined concentrations to assure the following final number of spermatozoa inside the straws: 15, 25, 35, 55, and 75 × 10^6^ sperm/straw, respectively (Figure 1; Table 1). Thus, each semen sample was processed and cryopreserved using the freezing technique as outlined by Iaffaldano et al. [7]. To prepare them for freezing, the semen samples that were prediluted with TGC were cooled at 5 °C for 90 min [9]. After cooling, they were diluted to a ratio of 1:1 (*v*:*v*) with a freezing extender composed of TCG containing 16% DMSO, as a permeable CPA, and 0.1 mol/L sucrose, as a non-permeable CPA. The diluted semen was aspirated into 0.25 mL plastic straws, equilibrated at 5 °C for 45 min (equilibration time), and frozen by exposure to liquid nitrogen vapor 5 cm above the liquid nitrogen surface (temperature was approximately −125/−130 °C) for 10 min. Finally, the straws were plunged into liquid nitrogen for storage at −196 °C. Five different straw colors were used to identify the different treatments (Figure 1). In total, 480 straws were frozen (12 × 5 sperm concentrations × 8 pool). The semen samples were thawed via the immersion of the straws into a water bath at 50 °C for 10 s. For each treatment, two straws per pool were thawed (total 80), and the analyses were carried out in duplicate, immediately after thawing and after 30 min.

### 2.5. Sperm Quality Assessment

The evaluation of semen quality during the cryopreservation process was conducted in five steps after: (1) dilution of the fresh semen, (2) the cooling process, (3) the equilibration phase, (4) thawing occurring, and (5) 30 min of thawing. This comprehensive assessment aimed to evaluate the overall influence of varying sperm concentrations at distinct time intervals and to analyze how cryopreservation affected the sperm quality parameters throughout the entire procedure.

The sperm-related variables included the sperm motility (total, progressive, and kinetic parameters) and sperm membrane integrity.

The sperm motility was estimated using a computer-aided sperm analysis (CASA) system coupled with a phase contrast microscope (Nikon Eclipse model 50i; negative contrast, Firenze, Italy) using the Sperm Class Analyzer (SCA) software (version 4.0, Microptic S.L., Barcelona, Spain) [11]. At each time point, the aliquots of semen from the different treatments (five sperm concentrations) were extended in 0.9% NaCl to reach a sperm concentration of 50 × 10^6^/mL. Following a 5 min incubation at 37 °C, 3 μL of the prepared sample was loaded onto a 20-micron Leja slide (Leja Standard Count, Nieuw Vennep, the Netherlands), prewarmed on a lab heating plate, and examined under the microscope at 100× total magnification. The following parameters were recorded: total motility (TM, %), progressive motility (PM, %), curvilinear velocity (VCL, (μm/s)), straight-line velocity (VSL, (μm/s)), average path velocity (VAP, (μm/s)), linearity (LIN, (%)), straightness (STR, (%)), wobble (WOB, (%)), amplitude of lateral head displacement (ALH, (μm)), and beat cross frequency (BCF, (Hz)). At least 500 sperm for each sample were observed considering three microscopic fields. The setting used is specifically for rabbit semen, provided by the SCA software, and the configurations used are the following: frame rate (fps) = 25, static (µm/sec) < 10; slow–medium (µm/sec) = 25, rapid (µm/sec) > 50, progressive (STR) > 70, connectivity (pixels) = 13.

The sperm membrane integrity (SMI) was measured using the Muse^®^ Cell Analyzer (Luminex Co., 12212 Technology Blvd Suite 130, Austin, TX 78727, USA) following the manufacturer’s protocol, in accordance with the methodology reported in Rusco et al. [11]. To perform the analysis, the semen samples were first diluted in phosphate-buffered saline (PBS) to achieve a concentration ranging from 1 × 10^5^ to 1 × 10^6^ spermatozoa/mL. Then, 20 μL of this suspension was combined with 780 μL (dilution factor 1:40) of the Muse Count and Viability Kit^®^ Reagent in an Eppendorf tube (Luminex Co.). The mixture was then incubated for 5 min at room temperature in the absence of light. Subsequently, the Eppendorf tubes were analyzed using flow cytometry. The accompanying software module performed calculations and presented the data in two dot plots: (1) nucleated cells—a membrane-permeant DNA staining dye that stained all cells with a nucleus (this plot functions to identify cells with a nucleus from debris and non-nucleated cells); (2) viability—a DNA-binding dye stained cells that had lost their membrane integrity and allowed the dye to stain the nucleus of dead and dying cells. This parameter discriminates viable (live cells that do not stain) from non-viable (dead or dying cells that stain) cells.

### 2.6. Statistical Analysis

Before analysis, the data were checked for normal distribution using the Shapiro–Wilk test. All the data were presented as mean ± standard error of the mean.

To compare the different treatments, we used a randomized block design in a five-by-five factorial arrangement: five sperm concentrations (15–25–35–55–75 × 10^6^ spz/straw) × five time points (fresh, cooled, equilibrated, thawed, and 30 min post-thawing), with eight replicates per treatment.

The sperm variables (CASA motility parameters and sperm membrane integrity) were compared among the treatments using a two-way ANOVA. Duncan’s comparison test was used to distinguish significantly different means. A generalized linear model (GLM) procedure was then used to determine the fixed effects of sperm concentrations, time points, and their interactions with the sperm quality variables. Significance was set at *p* < 0.05. All statistical tests were conducted using the software package SPSS (IBM SPSS Statistics 23.0 for Windows, 2020; SPSS, Chicago, IL, USA).

## 3. Results

The overall quality of the fresh semen in each pool was assessed before dilution, yielding positive results with the following average values: TM was 91.7 ± 2.6; PM 68.2 ± 4.2; SMI 90.4 ± 2.7; and the sperm concentration was 732.5 ± 158.6 × 10^6^ spermatozoa/mL. The Appendix A provide details on the quality of each semen pool, with the average values presented as mean ± standard deviation (Appendix A).

Table 2 provides information on the fixed effects of the sperm concentration, time point, and their interactions on the various sperm parameters evaluated. This table elucidates the significance of these factors in relation to the examined sperm variables. The key findings encompass the following: (1) a significant effect of sperm concentration was observed for TM, PM, VCL, STR, and ALH; (2) a significant effect of the time point on all sperm parameters was measured; (3) the effects of the interactions between the sperm concentration and time point were found to be significant for all sperm traits except for STR, LIN, and SMI.

### 3.1. Total and Progressive Motility Analysis

Figure 2 and Figure 3 show the outcomes related to TM and PM. No significant effect of the sperm concentration was observed for the TM in fresh semen, whilst higher values of PM were found in fresh semen diluted at the final concentration into straws of 75 × 10^6^ compared with the concentration of 15 × 10^6^ (*p* < 0.05).

There was a significant influence of the sperm concentration on both the TM and PM in the cooled semen. Specifically, significantly higher values of TM were observed when the sperm concentration was set at 15 and 25 × 10^6^ sperm/straw compared to the other sperm concentrations (*p* < 0.05). Better values (*p* < 0.05) for PM were recorded for the concentrations of 15, 25, and 35 × 10^6^ in respect to other concentrations; however, it is worth noting that no significant differences were found among them.

At the time point of equilibration, the highest values for both variables were obtained when the sperm concentration was 15 × 10^6^ with respect to the other concentrations. Moving on to the analysis of the motility parameters post-thawing, a higher TM (*p* < 0.05) was observed at the concentration of 25 × 10^6^ sperm/straw (49.9 ± 2.5%) in comparison to the concentrations of 15, 55, and 75 × 10^6^. Likewise, also, the values of PM were better (*p* < 0.05) at the concentration of 25 × 10^6^ sperm/straw (19.7 ± 1.4%) with respect to 15 and 75 × 10^6^.

As we expected, a gradual decline in both the TM and PM was observed across the various time points. Notably, for the TM, there were no significant differences recorded between the fresh and cooled semen for each final sperm concentration. However, after the equilibration phase, the value of these parameters showed a significant decrease across all sperm concentrations. After thawing, the TM values were remarkably higher in the samples evaluated immediately after thawing compared to those assessed 30 min post-thawing in all sperm concentrations (*p* < 0.05), except for in the 75 × 10^6^ concentration.

A similar trend was observed for the PM. In this regard, a significant decline between the fresh and cooled semen for the 55 and 75 × 10^6^ concentrations were noted. Transitioning from the cooling phase to the equilibration phase, a marked decrease (*p* < 0.05) in PM was consistently observed across all sperm concentrations.

Following the thawing process, akin to the trend seen in TM, higher PM values were observed in the semen analyzed immediately after thawing at each sperm concentration (*p* < 0.05), except for in the 75 × 10^6^ concentration, where the values remained nearly constant.

### 3.2. Other CASA Variables

The results related to the kinetic parameters are shown in Table 3. No significant effect of the sperm concentration was observed on the majority of these parameters in fresh semen, with the exception of LIN and WOB, where higher values were observed at a concentration of 55 × 10^6^ compared to 25 and 35 × 10^6^ (*p* < 0.05). Additionally, the lowest values of BCF were registered at the concentration of 15 × 10^6^ (*p* < 0.05).

The concentration exerted a significant influence on the following parameters in cooled semen—VCL, VAP, VSL, LIN, WOB, and ALH—with the concentration of 35 × 10^6^ generally resulting in the highest values.

At the time point of equilibration, the semen diluted at a final concentration of 15 × 10^6^/straw displayed significantly higher values in VCL, VAP, VSL, ALH, and BCF. This was in comparison to all other concentrations, apart from concentration 25 × 10^6^ for VAP and the concentrations of 25 and 35 × 10^6^ for VSL and BCF.

In the thawed semen, similar values for all parameters were observed among the concentrations 25, 35, 55, and 75 × 10^6^. In contrast, the concentration of 15 × 10^6^ returned significantly lower values across all kinetic parameters compared to the other concentrations, with the exclusion of WOB.

Thirty minutes after thawing, the concentrations of 35, 55, and 75 × 10^6^ had similar VAP, VSL, STR, LIN, WOB, and BCF values. These outcomes were significantly higher compared to those found in the concentrations of 15 and 25 × 10^6^. In accordance with what was observed for TM and PM, there was a gradual decrement in the kinetic parameters across the various time points. The differences were significant during the transition from cooled semen to equilibrated semen for VCL, VAP, VSL, ALH, and BCF in concentrations from 25 to 75 × 10^6^. In general, a significant decline was also registered between the equilibration time point and thawed semen. It should be emphasized that the values for all parameters were similar between the semen evaluated immediately after thawing and 30 min later.

### 3.3. Sperm Membrane Integrity

The results on SMI are depicted in Figure 4, and it is noteworthy that no significant effect of sperm concentration was observed at any of the time points assessed.

When analyzing the evolution of the freezing process, it was observed that there were no significant differences in the SMI among the fresh, cooled, and equilibrated semen at the concentrations of 15, 25, and 35 × 10^6^. The SMI values for the semen cooled and equilibrated at a final concentration of 75 × 10^6^ showed a significant decrement compared to those of fresh semen. As we moved from the equilibration step to post-thawing, a significant reduction was detected across all sperm concentrations tested. Nevertheless, the SMI values remained unchanged immediately after thawing or 30 min post-thawing for all sperm concentrations considered except for that of 15 × 10^6^.

## 4. Discussion

This study aimed to pinpoint the optimal sperm concentration in straws for the cryopreservation of rabbit semen, with an overarching objective of refining and standardizing the freezing protocol developed in our group’s research over the last decade [7,8,9,10,11]. Moreover, the discovery of the most suitable concentration for freezing rabbit semen marks an innovative stride, addressing an aspect that has received relatively limited attention within this field of research up until now. To the best of our understanding, this constitutes a pioneering and comprehensive report exploring the influence of the final sperm concentration in straws on the post-thaw quality of rabbit semen while maintaining a constant CPA concentration.

In this regard, most of the protocols established for semen cryopreservation in this species are based on dilution, employing fixed sperm-to-extender ratios, ranging from 1:1 to 1:10 [1,4,7,8,13,14,15,16,17,18,19,20,21,22]. The effect of the sperm concentration on the post-thaw quality of semen has been explored in different species, mainly in fish [23,24], bulls [25], horses [26,27], rams [28], and donkeys [29]. In a study carried out by Castellini et al. [30], they evaluated the impact of various sperm concentrations in the straws (10, 25, and 50 × 10^6^), and their results showed that the sperm concentration did not have a significant effect on the post-thaw sperm motility traits. Notably, the straws’ volume was different, and the extender composition (CPAs and concentrations) and the protocols were not provided and therefore not comparable with our experiment.

To gain a comprehensive understanding of the influence of the sperm concentration on rabbit semen quality, we conducted a thorough evaluation at various stages throughout the cryopreservation process. The obtained outcomes unveiled a significant effect of the sperm concentration on a range of sperm parameters (TM, PM, VCL, STR, and ALH) considering the entire cryopreservation process. Moreover, the effect of the time point showed a significant impact on all assessed sperm parameters, signifying a progressive decline in these variables as the cryopreservation process advanced through its stages (e.g., fresh, cooling, equilibration, post-thawing). This pattern aligns seamlessly with our previous findings [7,8,9,10,11]. However, when measuring the interaction between the sperm concentration and time point, we observed significant effects for the motility sperm traits. This suggests that the combined influence of the sperm concentration and the specific time point in the cryopreservation process had a notable impact on a wide range of sperm parameters. In this regard, the effect of the sperm concentration on freshly diluted semen was limited; nevertheless, it became increasingly pronounced after the cooling phase. In particular, the semen that underwent cooling and equilibration at a final concentration of 15 × 10^6^ spermatozoa per straw exhibited higher values for TM, PM, and the majority of the kinetic motility parameters. However, unexpectedly, this enhancement was not sustained after thawing, as the sample with the lowest sperm concentration showed a decline in quality compared to the other concentrations.

One plausible interpretation of this result is that the spermatozoa diluted at a concentration of 15 × 10^6^ had high availability of energy substrates during the cooling and equilibration period due to a greater extender-to-spermatozoa ratio. Conversely, despite the CPA effectively dehydrating the sperm cells during the equilibration phase, this lower concentration seems to promote, after freezing, the formation of greater number of ice crystals in the extracellular environment, as the aqueous portion appeared to be more substantial compared to the other concentrations, ultimately compromising the post-thawed semen quality. It is known that the sperm freezing process involves the development of extracellular ice crystals, which are formed from the surrounding water during cryopreservation [31]. These ice crystals elevate the solute concentration in the extracellular medium and generate an osmotic gradient across the sperm membrane [32,33]. The osmotic stress and physical harm caused by the formation of these external ice crystals result in membrane damage and ultimately lead to the death of the sperm cells [34]. This phenomenon is likely to be more pronounced at a concentration of 15 × 10^6^. Another plausible explanation could be due to the excessive dilution of the seminal plasma proteins. In this regard, Moreira et al. [35] showed that in mammals, the seminal plasma protein composition is associated with different functions in the sperm cells, such as motility, acrosomal reaction, freezability, cryoresilience, and fertilization events. For these reasons, the lowest concentration of 15 × 10^6^ implies reduced interactions among the plasma proteins and spermatozoa due to the high dilution, negatively influencing the post-thawing sperm quality.

After thawing, the effect of the sperm concentration was markedly evident in respect to the samples evaluated 30 min post-thawing: we observed that the TM and PM increased linearly with the sperm concentration up to that of 25 and 35 × 10^6^, after which a decline in these parameters was observed, with the result that the semen diluted at the highest concentration (75 × 10^6^) showed the worst values of TM and PM. However, this pattern did not hold true for the kinetic parameters, as consistently lower values post-thaw were observed in the lowest concentration (15 × 10^6^), while the other concentrations exhibited values that were nearly similar to each other. Our findings align with other research conducted in various species, where higher concentrations of sperm in the straws were linked to a consistent decrease in the percentages of total and progressive motile spermatozoa. In the case of horses, there is a consensus that a higher sperm concentration results in a lower proportion of motile spermatozoa after thawing [29].

For instance, Heitland et al. [26] investigated the impact of sperm concentrations in horses with a concentration range from 20 to 1600 × 10^6^ sperm/mL before freezing on the TM and PM. They observed a significant reduction in both parameters with the use of a higher sperm concentration. Similarly, in a study by Crockett et al. [27], higher post-thaw motility was reported after cryopreservation with concentrations of 50 and 250 × 10^6^ sperm/mL, whereas a significantly lower value was recorded at a higher concentration of 500 × 10^6^ sperm/mL. A decline in the TM and PM of cryopreserved semen with a higher pre-freezing sperm concentration has also been documented in dogs [36], rams [28,37,38], and fish [23,24]. Despite the fact that several studies have documented the impact of increased concentrations before freezing on the post-thaw quality, only a limited number of hypotheses have been proposed to elucidate this phenomenon.

In the realm of semen cryopreservation, there is a prevailing consensus that CPAs work by lowering the temperature at which sperm are exposed to a critical salt concentration [39]. Consequently, a decreased amount of CPAs available per cell may result in a diminished cryoprotective capacity. As reported by Contri et al. [29], this hypothesis gains support from the evident dose-dependent reduction in the cryoprotective effectiveness of the freezing extender at a higher sperm concentration. Furthermore, Nynca et al. [24] highlighted that the exact cause for the reduction in post-thaw sperm quality at high straw concentrations in trout remains elusive. Expanding upon the insights of Lahnsteiner [40], they propose that the increased number of sperm within each straw could potentially result in cellular compression due to the restricted intercellular space, consequently diminishing the post-thaw sperm quality.

Unexpectedly, throughout all time intervals examined, the sperm concentration displayed no effect on the SMI; however, after thawing, slightly higher values were noted in the spermatozoa frozen at the final concentration of 25 and 35 × 10^6^. This finding implies that, in the context of SMI, the choice of sperm concentration may hold less critical during the freezing process.

Another noteworthy finding of our study was the significant decrease in the post-thaw semen quality following a 30 min incubation at 37 °C; this was clearly noticeable in all concentrations except for 75 × 10^6^. Therefore, this finding implies that in the artificial insemination process, it is crucial to use thawed semen promptly, and at the latest within 30 min. This aligns with the findings in boars, where artificial insemination should be carried out within 10–30 min after thawing [41,42], and in bulls, where artificial insemination with frozen semen is recommended to take place even sooner than after 10 min [43].

In summary, the optimal sperm quality immediately after thawing occurs when semen is diluted and frozen at a final sperm concentration in straws of 25 and 35 × 10^6^. These concentrations strike the right balance between the sperm count and the available P-CPA and NP-CPA. However, 30 min after post-thawing, the concentration effect was annulled, eliminating the weaker sperm population, and leaving the more cryoresistant ones. Notably, two distinct sperm populations may coexist: a resilient group maintaining consistent total sperm motility across all concentrations 30 min post-thawing, and a less resistant group, more prominent in the 25 and 35 × 10^6^ concentration immediately after thawing. 

## 5. Conclusions

We noted that the mid-range sperm concentrations (25 and 35 × 10^6^/straw) exhibited a superior sperm quality immediately after thawing compared to the other concentrations. However, it is noteworthy that, after a 30 min post-thawing period, the semen quality no longer showed the discernible influence of the sperm concentration. To achieve a more nuanced understanding and a comprehensive assessment of the impact of the sperm concentration, further investigations are warranted. In pursuit of this, we have scheduled in vivo tests within artificial insemination practices, aiming to delve deeper into the intricate dynamics of sperm concentration and its lasting effects on semen quality. These results provide valuable guidance to the scientific community in unifying and improving the freezing technique for rabbit semen.

Standardizing the sperm concentration in each straw is essential to minimize result variability and accurately determine the number of sperm received by each doe during the AI procedures, guaranteeing the reproducibility and accuracy of the AI technique, which is particularly crucial for successful rabbit population breeding management. In the context of cryobank purposes, precise sperm concentration controls are essential to ensure the genetic integrity and viability of the stored genetic material. By adhering to standardized sperm concentrations, cryobanks can confidently preserve the genetic resources of rabbit populations and provide a valuable resource for future breeding and research efforts.

## Figures and Tables

**Figure 1 vetsci-11-00009-f001:**
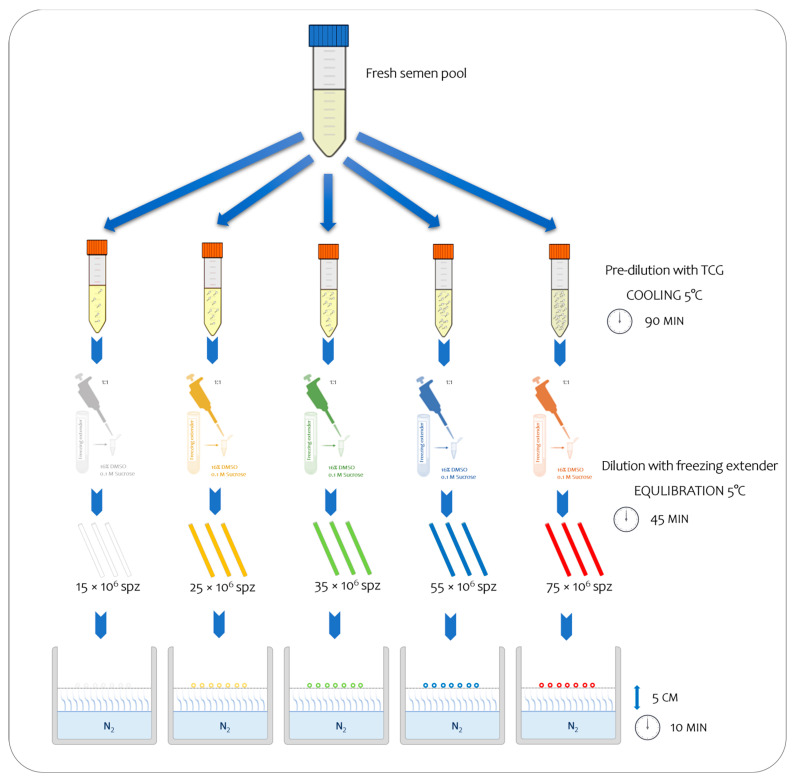
Experimental study design, enabling the observation of sequential steps from fresh to frozen semen at five distinct final sperm concentrations.

**Figure 2 vetsci-11-00009-f002:**
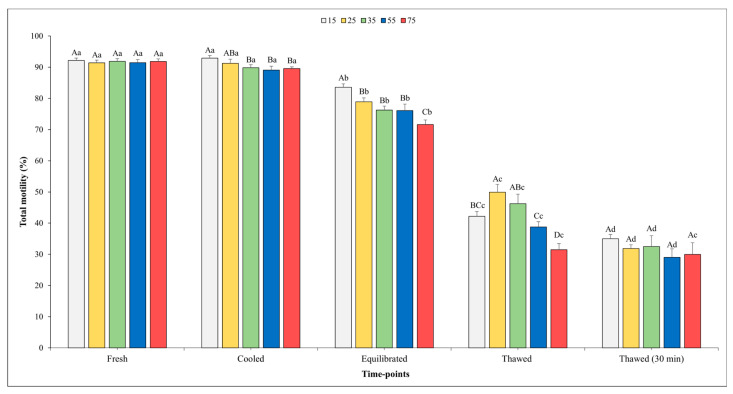
Outcomes of total motility in relation to different sperm concentrations/straws throughout different time points of cryopreservation process (fresh, cooled, equilibrated, and thawed). Different uppercase letters (A–D) indicate statistical significances at *p* < 0.05 among different sperm concentrations within each time point; different lowercase letters (a–d) indicate statistical significances at *p* < 0.05 among different time points within each sperm concentration. Sperm concentrations (15, 25, 35, 55, and 75) are expressed as n° spz × 10^6^/straw.

**Figure 3 vetsci-11-00009-f003:**
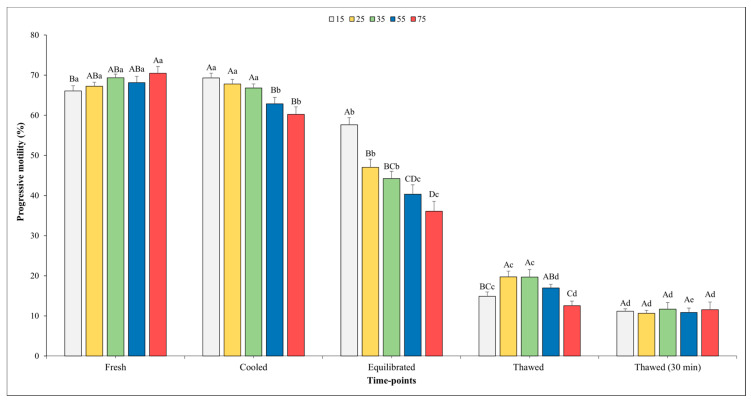
Outcomes of progressive motility in relation to different sperm concentrations/straw throughout different time points of cryopreservation process (fresh, cooled, equilibrated, and thawed). Different uppercase letters (A–D) indicate statistical significances at *p* < 0.05 among different sperm concentrations within each time point; different lowercase letters (a–e) indicate statistical significances at *p* < 0.05 among different time points within each sperm concentration. Sperm concentrations (15, 25, 35, 55, and 75) are expressed as n° spz × 10^6^/straw.

**Figure 4 vetsci-11-00009-f004:**
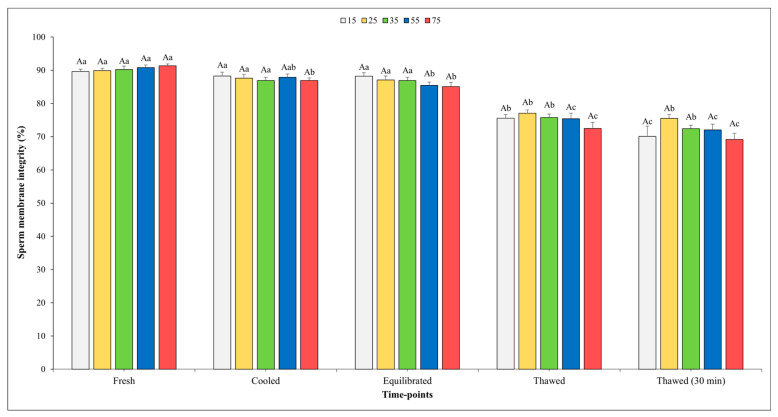
Outcomes of sperm membrane integrity in relation to different sperm concentrations/straw throughout different time points of cryopreservation process (fresh, cooled, equilibrated, and thawed). Uppercase letter (A) indicates statistical significances at *p* < 0.05 among different sperm concentrations at each time point; different lowercase letters (a–c) indicate statistical significances at *p* < 0.05 among different time points within each sperm concentration. Sperm concentrations (15, 25, 35, 55, and 75) are expressed as n° spz × 10^6^/straw.

**Table 1 vetsci-11-00009-t001:** Example of dilution of fresh rabbit semen (650 × 10^6^ spermatozoa mL^−1^) with TCG and freezing extender to reach final sperm concentrations in the straws (15, 25, 35, 55 and 75 × 10^6^ spermatozoa).

	First Dilution with TCG				Second Dilution with F.E.			
Initial Fresh Sperm Concentration (×10^6^ mL^−1^)	Sperm Concentration after Pre-Dilution with TCG (×10^6^ mL^−1^)	Dilution Rate	Semen Volume (mL)	TCG Volume (mL)	F.E. Volume (mL)	Final Volume (mL)	Sperm Concentration after Dilution with F.E. 1:1 (×10^6^ mL^−1^)	Sperm Concentration Per Straw (×10^6^)
650	120	5.42	0.46	2.04	2.50	5.00	60	15
650	200	3.25	0.77	1.73	2.50	5.00	100	25
650	280	2.32	1.08	1.42	2.50	5.00	140	35
650	440	1.48	1.69	0.81	2.50	5.00	220	55
650	600	1.08	2.31	0.19	2.50	5.00	300	75

TCG: tris, citric acid, glucose; F.E.: freezing extender composed of TCG, 16% DMSO, and 0.1 M sucrose.

**Table 2 vetsci-11-00009-t002:** Effect of sperm concentrations, time points, and their interactions on sperm parameters as assessed using GLM analysis.

Sperm Variables	Concentration Effect	Time Point Effect	Concentration × Time Point Effect
TM	*p* = 0.000	*p* = 0.000	*p* = 0.000
PM	*p* = 0.000	*p* = 0.000	*p* = 0.000
VCL	*p* = 0.001	*p* = 0.000	*p* = 0.000
VAP	*p* = 0.302	*p* = 0.000	*p* = 0.000
VSL	*p* = 0.431	*p* = 0.000	*p* = 0.001
STR	*p* = 0.003	*p* = 0.000	*p* = 0.416
LIN	*p* = 0.054	*p* = 0.000	*p* = 0.186
WOB	*p* = 0.077	*p* = 0.000	*p* = 0.036
ALH	*p* = 0.000	*p* = 0.000	*p* = 0.000
BCF	*p* = 0.345	*p* = 0.000	*p* = 0.000
SMI	*p* = 0.088	*p* = 0.000	*p* = 0.430

TM (%), total motility; PM (%) progressive motility; VCL (µm/sec), curvilinear velocity; VAP (µm/sec), average path velocity; VSL (µm/sec), straight-line velocity; STR (%) (VSL/VAP × 100), straightness; LIN (%) (VSL/VCL × 100), linearity; WOB (%) (VAP/VCL × 100), wobble; ALH (µm), amplitude of lateral head displacement; BCF (Hz), beat cross frequency; SMI (%) sperm membrane integrity.

**Table 3 vetsci-11-00009-t003:** Effect of different straws’ sperm concentrations on kinetic parameters of rabbit semen during different time points (fresh, cooled, equilibrated, and thawed) of cryopreservation process.

		Time Points				
Sperm Variables	Sperm Concentration	Fresh	Cooled	Equilibrated	Thawed	Thawed (30 min)
VCL (µm/sec)	**15** **25** **35** **55** **75**	73.0 ± 2.5 ^Ab^ 75.4 ± 1.9 ^Aa^ 76.3 ± 1.5 ^Aa^ 75.1 ± 2.8 ^Aa^ 75.9 ± 2.1 ^Aa^	79.0 ± 2.1 ^Aa^ 78.8 ± 1.8 ^Aa^ 77.8 ± 2.6 ^Aa^ 74.2 ± 1.7 ^ABa^ 69.5 ± 2.4 ^Bb^	75.7 ± 2.4 ^Aab^ 64.5 ± 2.3 ^Bb^ 60.1 ± 2.0 ^BCb^ 55.4 ± 2.6 ^CDb^ 51.5 ± 2.3 ^Dc^	41.9 ± 1.3 ^Cc^ 44.6 ± 1.0 ^BCc^ 47.7 ± 1.4 ^ABc^ 49.2 ± 1.6 ^Ac^ 46.1 ± 1.8 ^ABCc^	36.9 ± 1.0 ^Ac^ 38.9 ± 1.2 ^Ad^ 40.4 ± 1.8 ^Ad^ 39.6 ± 1.3 ^Ad^ 40.0 ± 1.8 ^Ad^
VAP (µm/sec)	**15** **25** **35** **55** **75**	40.8 ± 2.4 ^Aa^ 38.2 ± 1.8 ^Aa^ 40.1 ± 1.9 ^Ab^ 45.5 ± 4.1 ^Aa^ 41.6 ± 1.8 ^Aa^	40.8 ± 2.2 ^ABa^ 39.4 ± 1.1 ^Ba^ 45.4 ± 2.3 ^Aa^ 37.8 ± 0.9 ^Bb^ 37.3 ± 1.3 ^Bb^	39.6 ± 2.3 ^Aa^ 34.7 ± 2.3 ^ABb^ 31.0 ± 1.9 ^BCc^ 28.5 ± 1.9 ^Cb^ 28.0 ± 1.9 ^Cc^	20.0 ± 0.5 ^Bb^ 21.5 ± 0.4 ^ABc^ 22.7 ± 0.7 ^Ad^ 22.8 ± 0.8 ^Acd^ 22.7 ± 0.8 ^Ad^	17.6 ± 0.7 ^Bb^ 18.0 ± 0.5 ^Bc^ 20.5 ± 0.9 ^Ad^ 19.1 ± 0.7 ^Ad^ 20.6 ± 0.9 ^Ad^
VSL (µm/sec)	**15** **25** **35** **55** **75**	27.1 ± 2.2 ^Aa^ 24.4 ± 1.3 ^Aa^ 26.2 ± 1.5 ^Ab^ 31.2 ± 3.9 ^Aa^ 26.7 ± 1.8 ^Aa^	26.0 ± 2.0 ^Ba^ 24.8 ± 0.9 ^Ba^ 30.8 ± 2.2 ^Aa^ 23.5 ± 0.9 ^Bb^ 23.6 ± 1.1 ^Ba^	27.3 ± 2.4 ^Aa^ 24.2 ± 2.5 ^ABa^ 21.3 ± 2.1 ^ABc^ 19.2 ± 1.9 ^Bb^ 19.5 ± 1.8 ^Bb^	10.1 ± 0.4 ^Bb^ 11.3 ± 0.5 ^ABb^ 12.2 ± 0.4 ^Ad^ 12.6 ± 0.5 ^Ac^ 12.6 ± 0.6 ^Ac^	10.2 ± 0.8 ^Cb^ 10.8 ± 0.5 ^BCb^ 13.3 ± 0.7 ^Ad^ 12.3 ± 0.5 ^ABc^ 13.1 ± 0.6 ^Ac^
STR (%)	**15** **25** **35** **55** **75**	61.8 ± 1.4 ^Aa^ 60.9 ± 1.2 ^Aa^ 61.0 ± 0.9 ^Aa^ 62.9 ± 1.7 ^Aa^ 61.7 ± 1.5 ^Aab^	59.5 ± 1.3 ^Aa^ 59.6 ± 0.8 ^Aa^ 63.4 ± 1.9 ^Aa^ 59.9 ± 1.0 ^Aa^ 61.1 ± 0.8 ^Ab^	62.8 ± 2.1 ^Aa^ 62.3 ± 2.0 ^Aa^ 62.8 ± 2.2 ^Aa^ 63.0 ± 1.7 ^Aa^ 64.8 ± 1.8 ^Aa^	45.5 ± 0.9 ^Cc^ 49.0 ± 1.0 ^Bb^ 50.0 ± 1.0 ^ABc^ 50.5 ± 0.6 ^ABc^ 51.8 ± 0.9 ^Ac^	50.1 ± 1.1 ^Bb^ 52.1 ± 1.1 ^Bb^ 54.5 ± 1.0 ^Ab^ 55.8 ± 1.0 ^Ab^ 54.5 ± 0.8 ^Ac^
LIN (%)	**15** **25** **35** **55** **75**	36.9 ± 2.4 ^ABa^ 32.7 ± 1.5 ^Bab^ 33.2 ± 1.4 ^ABab^ 39.4 ± 3.1 ^Aa^ 35.6 ± 1.8 ^ABa^	32.3 ± 1.9 ^Ba^ 31.4 ± 0.9 ^Bb^ 39.5 ± 2.8 ^Aa^ 32.3 ± 1.4 ^Bbc^ 35.3 ± 1.8 ^ABa^	35.8 ± 3.0 ^Aa^ 36.8 ± 3.1 ^Aa^ 35.7 ± 3.4 ^Aa^ 36.5 ± 2.6 ^Aab^ 39.7 ± 2.4 ^Aa^	22.6 ± 0.7 ^Cb^ 24.7 ± 0.8 ^ABc^ 24.8 ± 0.8 ^ABc^ 23.9 ± 0.6 ^BCd^ 26.5 ± 0.7 ^Ab^	24.8 ± 1.0 ^Bb^ 25.4 ± 0.8 ^Bc^ 28.9 ± 1.0 ^Ac^ 28.2 ± 1.0 ^Acd^ 29.2 ± 1.0 ^Ab^
WOB (%)	**15** **25** **35** **55** **75**	56.2 ± 2.3 ^ABa^ 51.3 ± 1.5 ^Bab^ 52.1 ± 1.7 ^Bb^ 58.9 ± 2.8 ^Aa^ 55.1 ± 1.5 ^ABa^	51.6 ± 1.8 ^Ba^ 50.5 ± 1.0 ^Babc^ 58.4 ± 2.4 ^Aa^ 51.8 ± 1.4 ^Bbc^ 54.8 ± 1.7 ^ABa^	53.1 ± 2.6 ^Aa^ 54.5 ± 2.7 ^Aa^ 52.7 ± 2.9 ^Ab^ 54.1 ± 2.3 ^Aab^ 56.7 ± 2.0 ^Aa^	47.6 ± 0.7 ^Ab^ 48.5 ± 0.7 ^Abc^ 47.6 ± 0.6 ^Ab^ 45.7 ± 0.6 ^Bd^ 49.1 ± 0.6 ^Ab^	46.6 ± 1.0 ^Bb^ 46.2 ± 0.8 ^Bc^ 49.7 ± 0.9 ^Ab^ 47.9 ± 1.1 ^ABcd^ 50.2 ± 1.1 ^Ab^
ALH (µm)	**15** **25** **35** **55** **75**	3.3 ± 0.2 ^Aa^ 3.5 ± 0.1 ^Aa^ 3.3 ± 0.1 ^Aa^ 3.1 ± 0.1 ^Aa^ 3.4 ± 0.1 ^Aa^	3.7 ± 0.1 ^Aa^ 3.7 ± 0.1 ^Aa^ 3.4 ± 0.2 ^ABa^ 3.5 ± 0.1 ^ABa^ 3.2 ± 0.2 ^Ba^	3.4 ± 0.2 ^Aa^ 3.0 ± 0.1 ^Bb^ 2.9 ± 0.1 ^Bb^ 2.7 ± 0.1 ^Cc^ 2.4 ± 0.1 ^Cb^	2.3 ± 0.1 ^Bb^ 2.5 ± 0.1 ^ABc^ 2.5 ± 0.1 ^ABc^ 2.6 ± 0.1 ^Ac^ 2.4 ± 0.1 ^ABb^	2.1 ± 0.1 ^Ab^ 2.1 ± 0.2 ^Ad^ 2.1 ± 0.1 ^Ad^ 2.1 ± 0.1 ^Ad^ 2.1 ± 0.2 ^Ac^
BCF (Hz)	**15** **25** **35** **55** **75**	6.9 ± 0.2 ^Ba^ 7.6 ± 0.2 ^Aa^ 7.7 ± 0.2 ^Aa^ 7.4 ± 0.1 ^ABa^ 7.9 ± 0.2 ^Aa^	7.0 ± 0.2 ^Aa^ 7.2 ± 0.2 ^Aa^ 7.1 ± 0.1 ^Ab^ 7.1 ± 0.2 ^Aa^ 6.9 ± 0.2 ^Ab^	6.8 ± 0.3 ^Aa^ 6.1 ± 0.3 ^ABb^ 6.1 ± 0.2 ^ABc^ 5.7 ± 0.2 ^Bb^ 5.8 ± 0.2 ^Bc^	3.6 ± 0.1 ^Bb^ 4.0 ± 0.1 ^ABc^ 4.1 ± 0.1 ^Ad^ 4.3 ± 0.1 ^Ac^ 4.1 ± 0.2 ^Ad^	3.7 ± 0.2 ^Bb^ 3.6 ± 0.1 ^Bc^ 4.1 ± 0.2 ^Ad^ 4.4 ± 0.1 ^Ac^ 4.2 ± 0.1 ^Ad^

Different superscripts (A–D) within the same column indicate a significant effect of sperm concentration within each time point (*p* < 0.05). Different superscripts (a–d) within the same row indicate a significant effect of time points within each sperm concentration (*p* < 0.05). Sperm concentrations (15, 25, 35, 55, and 75) are expressed as n° spz × 10^6^/straw. VCL (µm/sec), curvilinear velocity; VAP (µm/sec), average path velocity; VSL (µm/sec), straight-line velocity; STR (%) (VSL/VAP × 100), straightness; LIN (%) (VSL/VCL × 100), linearity; WOB (%) (VAP/VCL × 100), wobble; ALH (µm), amplitude of lateral head displacement; BCF (Hz), beat cross frequency.

## Data Availability

No new data were created or analyzed in this study. Data sharing is not applicable to this article.

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
