# Peer review of "Cryopreserving Rabbit Semen: Impact of Varying Sperm Concentrations on Quality and the Standardization of Protocol"

_vetsci, 2023, doi:10.3390/vetsci11010009_

Round 1
Reviewer 1 Report
Comments and Suggestions for Authors
This is an interesting manuscript that evaluated the impact of varying sperm concentrations on quality and standardization of protocol for rabbit semen cryopreservation. It is well written in general; only some minor points should be adjusted.
Abstract - Authors should include some more details related to methodology, if possible, as number of individuals used per semen pool and number of pools used, methods for analyzing sperm motility and membrane integrity. Also, numeric results would be welcome.
Introduction -
- Authors begin introduction with the statement "The limited survival rate of rabbit sperm following cryopreservation represents a significant challenge that hinders the widespread adoption of frozen semen in artificial insemination (AI) programs ". In all the text, authors highlight the need for improving or standardizing protocols for rabbit semen cryopreservation, but they do not report the average percentage of sperm survival after cryopreservation in the species usually achieved. Please, include these data to support the needs for protocol improvement.
- Why did you choose to test sperm concentrations varying from 15 to 75 million? What are the literature references that support this choice? What is the sperm concentration usually found for a rabbit ejaculate? What is the sperm concentration usually used for Artificial insemination? Please, include these statements on your introduction.
Material and Methods
- In CASA analysis, please indicate the settings used. Were they specific for rabbit semen?
- Please, provide reference for the validation of this membrane integrity analysis for rabbit sperm.
Results - Even if the tables and figures are welcome, the text related to the results is a few unorganized and sometimes difficult to be comprehended. I suggest authors to separe the text in different subtopics, emphasizing: 1. Total and progressive motility analyisis; 2. Other CASA variables; 3. Sperm membrane integrity
Discussion - It is well designed in general. In my opinion, authors should state about the presence of seminal plasma and its amount into the samples. It is known for some species that it is necessary for a good cryopreservation. Therefore, an excessive dilution as that observed at a 15-million concentration would reduce the interactions among plasma proteins and sperm. Please see: Moreira SSJ, Lago AEA, Moura AAA, Silva AR. Impact of Seminal Plasma Composition on Sperm Freezability in Wild Mammals: A Review. Biopreserv Biobank. 2022 Feb;20(1):90-96. doi: 10.1089/bio.2021.0026. Epub 2021 Nov 2. PMID: 34726507.
Reviewer 2 Report
Comments and Suggestions for Authors
I congratulate the authors on the presentation of a thorough paper investigation the, often neglected area of, optimization of sample quality and packaging for biobanking and end-use application of assisted reproduction.
I have only a few minor suggestions that I ask you consider for the benefit of the reader.
1/ The methodology states you collected ejaculates to create 8 pools for the experiment, but it is unclear if those initial pools were diluted to 650x10^6/ml before the dilution in TCG for the sperm concentration tests. Inclusion of the range of fresh parameters would be interesting, perhaps as supplementary material.
2/ Seminal plasma is known to impact the success of sperm cold storage and cryopreservation across a range of taxa including rabbits, yet there is no discussion of this in the manuscript. Could some of the effects you are seeing be attributed to decreasing seminal plasma (SP) concentration through dilution, rather than sperm conc,?
3/ Ln 399 - how do you know the ice crystals are larger when sperm concentration is low? A citation is needed here.
4/ It was somewhat confusing to read "time" as a significant factor, as the only time assessment is post-thaw at 0 and then 30 minutes. It would be more accurate to refer to the factor as "stage" or "step" effects unless specifically referring to the post-thaw stages.
5/ Was cryoprotectant washed off before post-thaw assessment? Could the post-thaw effects be due to CPA dilution as higher concentration sperm samples would require more dilution to reach the 50x10^6 assessment concentration?
Comments on the Quality of English LanguageThe majority of the manuscript is well written and easy to follow.
The final paragraph of the introduction (ln 85-92), however, is confusing and needs grammatical changes. suggested edits:
"In this study, in order to find a standardized freezing protocol for this species, our main goal was to assess the impact of varying sperm concentrations within straws (15, 25, 86 35, 55 and 75 × 106) through on the quality of cryopreserved semen by evaluating the sperm motility parameters and sperm membrane integrity. However, a A comprehensive investigation of evaluating the effects of sperm concentration in across critical steps of the cryopreservation procedure, spanning from initial dilution with the extender, through to cooling, equilibration, and ultimately thawing was undertaken. This elucidated the understanding of the role of sperm concentration in the overall success of rabbit semen cryopreservation."
Reviewer 3 Report
Comments and Suggestions for Authors
Di Iorio et al. present novel data on the effects of sperm concentration on the motility and membrane integrity of rabbit spermatozoa. The experiment was well-designed and methods clearly described. Results were appropriately analyzed and discussed.
Specific Comments:
1) Lines 119-120: I would suggest, "...sperm concentration was assessed by spectrophotometric analysis."
2) Lines 131-132: Were the samples cooled to 5 C thru direct contact with 5 C air or were they cooled in a water bath that started at 20-25 C and slowly reached 5 C over 90 minutes?
3) Lin 138: I would suggest, "...straw colours..."
4) Lines 141-142: Samples were not further diluted after thawing and remained in the freezing media for all evaluations?
5) Lines 373-375: This sentence should be combined with the preceding paragraph.
6) Lines 376-395: These paragraphs could be combined into one.
7) Lines 403-404: I would suggest, "It is known that the sperm freezing process involves the development of extracellular ice crystals, which..."
8) Lines 422-433: These paragraph should be combined with the preceding paragraph.
9) Lines 434-445: These paragraphs should be combined.
Comments on the Quality of English Language
See above.
Reviewer 4 Report
Comments and Suggestions for Authors
Thank you for the work done on rabbit sperm concentration during the process of cryopreservation.
Results. Please would you reorganize the results in the same way as described in material and methods starting from sperm motility and kinematic parameters to sperm membrane integrity assessment.
How can you explain the results of the brutal decrease in sperm total motility parameters from 90% to 30% while sperm membrane integrity did not change after the thawing process and knowing that the thawing process is harmful to sperm membrane?
In the discussion, you mention a decline in sperm quality but it's not true since in this study you have only results of sperm motility. ex line 381, 394...etc
At which concentration did you analyze your samples in the CASA system? This parameter is very important since sperm concentration should be the same when analyzing motility.
Why 25 and 35 concentration is better than the other since after 30min thawing sperm total motility and progressivity was the same for all concentration....?
Please review this point and try to give more sense and explanation to the results even if it would be better to have more quality parameters such as ROS creation, mitochondrial status ...etc.
Thank you,
Round 2
Reviewer 4 Report
Comments and Suggestions for Authors
Accept manuscript as it is